# FAST 3D ACOUSTIC SCATTERING VIA DISCRETE LAPLACIAN BASED IMPLICIT FUNCTION ENCODERS

## ABSTRACT

Acoustic properties of objects corresponding to scattering characteristics are frequently used for 3D audio content creation, environmental acoustic effects, localization and acoustic scene analysis, etc. The numeric solvers used to compute these acoustic properties are too slow for interactive applications. We present a novel geometric deep learning algorithm based on discrete-laplacian and implicit encoders to compute these characteristics for general 3D objects at interactive rates. We use a point cloud approximation of each object, and each point is encoded in a high-dimensional latent space. Our multi-layer network can accurately estimate these acoustic properties for arbitrary topologies and takes less than 1ms per object on a NVIDIA GeForce RTX 2080 Ti GPU. We also prove that our learning method is permutation and rotation invariant and demonstrate high accuracy on objects that are quite different from the training data. We highlight its application to generating environmental acoustic effects in dynamic environments.

## 1 INTRODUCTION

Acoustic scattering corresponds to the disturbance of a given incident sound field due to an object's shape and surface properties. It can be regarded as one of the fundamental characteristics of an object. The effect of scattering can be expressed in terms of a scattered sound field, which satisfies Sommerfield's radiation condition. There is considerable work on modeling and measuring the acoustic scattering properties in physics and acoustics and these characteristics are widely used for sound rendering in games and virtual reality (Mehra et al., 2015; Rungta et al., 2018), noise analysis in indoor scenes (Morales & Manocha, 2018), acoustic modeling of concert halls (Shtrepi et al., 2015), non-line-of-sight (NLOS) imaging (Lindell et al., 2019), understanding room shapes (Dokmanić et al., 2013), receiver placement (Morales et al., 2019), robot sound source localization (An et al., 2019), 3D mapping (Kim et al., 2020), audio-visual analysis (Sterling et al., 2018), etc.

Acoustic scattering of objects can be modeled accurately using the theory of wave acoustics (Kuttruff, 2016). The scattering characteristics of objects are widely used for sound propagation, which reduces to solving the wave equation in large environments. Given a sound source location and its vibration patterns, acoustic simulation methods are used to predict the perceived sound at another specified location considering the medium it passes through and objects/boundaries it interacts with. While the wave behavior of sound is well understood in physics, it is much more difficult to compute acoustic scattering and sound propagation effects, especially for higher frequencies. Even with state-of-the-art acoustic wave solvers, it can take from hours to days to solve a moderately modeled room environment on a powerful workstation. One of the contributing factors to this difficulty is that wave behaviors are frequency dependent, so many frequency bands need to be analyzed separately.

Current methods for computing the acoustic scattering characteristics can use numeric solvers like boundary-element methods (BEM). However, their complexity increases as a cubic function of the frequencies and most current implementations are limited to static scenes or environments. No good or practical solutions are known to compute the acoustic scattering properties for dynamic environments or when objects move or undergo deformation.

**Main Results:** We present novel techniques based on geometric deep learning on differential co-ordinates to approximate the acoustic scattering properties of arbitrary objects. Our approach is general and makes no assumption about object's shape, genus, or rigidity. We approximate the

objects using point-clouds, and each point in the point cloud representation is encoded in a high-dimensional latent space. Moreover, the local surface shapes in the latent space are encoded using implicit surfaces. This enables us to handle arbitrary topology. Our network takes the point cloud as an input and outputs the spherical harmonic coefficients that represent the acoustic scattering field.

We present techniques to generate the ground truth data using an accurate wave-solver on a large geometric dataset. We have evaluated the performance on thousands of objects that are very different from the training database (with varying convexity, genus, shape, size, orientation) and observe high accuracy. We also perform an ablation study to highlight the benefits of our approach. The additional runtime overhead of estimating the scattering field from neural networks is less than 1ms per object on a NVIDIA GeForce RTX 2080 Ti GPU. We also prove that our learning method is permutation and rotation invariant, which is an important characteristic for accurate computation of acoustic scattering fields.

## 2 RELATED WORKS

### 2.1 WAVE-ACOUSTIC SIMULATION

Wave-acoustic simulation methods aim to solve the wave equation that governs the propagation and scattering of sound waves. Some conventionally used numeric methods include the finite-element method (Thompson, 2006), the boundary-element method (Wrobel & Kassab, 2003), and the finite-difference time domain (Botteldooren, 1995). The common requirement of these methods is the proper discretization of the problem domain (e.g., spatial resolution, time resolution), which means the time complexity of them will scale drastically with the simulation frequency. Recent works manage to speed up wave simulations by parallelized rectangular decomposition (Morales et al., 2015) or by using pre-computation structures (Mehra et al., 2015). An alternative way to use wave simulation results in real-time applications is to pre-compute a large amount of sound fields in a scene and compressing the results using perceptual metrics (Raghuvanshi & Snyder, 2014). While pre-computation methods can save significant runtime cost, they still require non-trivial pre-computation efforts for unseen scenarios, restricting their use cases.

### 2.2 LEARNING-BASED ACOUSTICS

Machine learning techniques have been widely applied in many popular computer science research areas. There is considerable work on developing machine learning methods for applications corresponding to audio-visual analysis (Zhang et al., 2017) and acoustic scene classification (Abeßer, 2020) (see Bianco et al. (2019) for a thorough list of work). In comparison, there are much fewer works studying the generation of acoustic data from a physical perspective. Therefore, methodologies from existing popular domains often do not directly apply to our problem of interest. Some more relevant works focus on estimating room acoustics parameters from recorded signals (Eaton et al., 2016; Genovese et al., 2019; Tsokaktsidis et al., 2019; Tang et al., 2020), which can help physically-based simulators to model real-world acoustics more faithfully. Pulkki & Svensson (2019) propose to use a neural network to model the acoustic scattering effect from rectangular plates without running simulation. This is closely related to our goal of bypassing expensive wave simulation while generating plausible sound, although we aim to model objects of more general shapes. Recently, Fan et al. (2020) train convolutional neural networks (CNNs) to learn to map planar sound fields induced by convex scatterers in 2D, which provides promising results. Motivated by the last method, we aim to develop networks that can deal with object geometry in 3D and extend the generality of this learning-based approach.

### 2.3 GEOMETRIC DEEP LEARNING AND SHAPE REPRESENTATION

There is considerable recent work on generating plausible shape representations for 3D data, including voxel-based (Zhou & Tuzel, 2018; Sindagi et al., 2019; Meng et al., 2019; Wu et al., 2015), point-based (Charles et al., 2017; Qi et al., 2017; Wang et al., 2019; Li et al., 2018a; Monti et al., 2017; Li et al., 2018b; Yi et al., 2017) and mesh-based (Hanocka et al., 2019) shape representations. This includes work on shape representation by learning implicit surfaces on point clouds (Smirnov et al., 2019), designing a mesh Laplacian for convolution (Tan et al., 2018), hierarchical graph

convolution on meshes (Mo et al., 2019), encoding signed distance functions for surface reconstruction (Park et al., 2019), etc. However, previous methods on point cloud shape representations learn by designing loss functions to constrain surface smoothness on global Cartesian coordinates. Such functions only provide spatial information of each point and lack information about local shape of the surface compared to the explicit discretization of the continuous Laplace-Beltrami operator and curvilinear integral (Do Carmo, 2016). Instead, we use point cloud based learning algorithms, which do not require mesh Laplacians for graph neural networks. This makes our approach applicable to all kind of dynamic objects, including changing topologies.

## 3 OUR APPROACH

### 3.1 OVERVIEW

Our goal is to directly predict the acoustic scattering field (ASF) resulting from an incident wave and a sound scatterer of known shape (described by a triangle mesh). A set of frequency dependent ASFs are visualized in Figure 1 on a disk plane. The two key steps are defining a compact and efficient way to encode the ASF, and designing a neural network architecture that can effectively exploit the 3D geometric information so as to establish the mapping between the object shape and the ASF. In this section, we provide necessary background on wave acoustics and explain the mechanism of our approach. Our notations are summarized in Table 1.

Table 1: Notation and symbols used in this section.

| | | | |
|---|---|---|---|
| $\mathbf{x}$ | 3D Cartesian coordinates. | $m, l$ | Order and degree of basis functions. |
| $v$ | acoustic particle velocity. | $Y_l^m(\theta, \phi)$ | Spherical harmonics basis function. |
| $\omega$ | Frequency of sound. | $k$ | Wavenumber. |
| $(r, \theta, \phi)$ | Spherical coordinates. | $c_l^m(\omega)$ | Spherical harmonics coefficients. |
| $c$ | Speed of sound in air. | $h_l(kr)$ | Hankel function. |

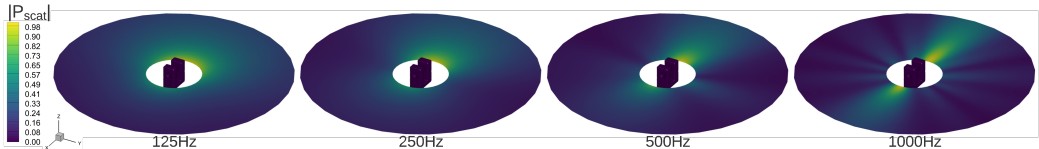

Figure 1: Acoustic scattering of the same object at four different frequencies, assuming a plane wave travelling to the $-x$ direction and a sound-hard boundary condition for the object/scatterer in the center. The scattering patterns are significantly different among frequency bands. Our learning method can compute them with high accuracy for arbitrary objects at interactive rates, including high-genus and deforming objects.

### 3.2 ACOUSTIC SCATTERING FIELDS

We are interested in knowing how the incident sound field will be scattered when it acts on an object surface. Let the sound pressure at location $\mathbf{x}$ and time $t$ be $P(\mathbf{x}, t)$, and the wave equation describes the propagation of sound:

$$(\nabla^2 - \frac{1}{c^2}\frac{\partial^2}{\partial t^2})P(\mathbf{x}, t) = 0, \tag{1}$$

where $c$ is the speed of sound, which is commonly assumed to be constant in non-large scale indoor environments. Transforming into the frequency domain, we obtain the homogeneous Helmholtz equation:

$$(\nabla^2 + k^2)p(\mathbf{x}, \omega) = 0, \tag{2}$$

where $\omega$ is the wave frequency and $k = \frac{\omega}{c}$ is the wavenumber. When a specific boundary condition is given, its solution in 3D spherical coordinates can be decomposed as

$$p(r, \theta, \phi, \omega) = \sum_{l=0}^{\infty} h_l(kr) \sum_{m=-l}^{+l} c_l^m(\omega) Y_l^m(\theta, \phi), \tag{3}$$

where $h_l(kr)$ is the Hankel function which encodes the radial pressure changes, $Y_l^m(\theta, \phi)$ is the spherical harmonics term which represents the angular part of the solution, and $c_l^m(\omega)$ is the spherical harmonics coefficients for frequency $\omega$. For a maximum order $L$, there are $(L+1)^2$ coefficients. Since the Hankel and the Spherical Harmonics functions are easy to evaluate, we only need to find the coefficients $c_l^m(\omega)$ to fully describe the acoustic scattering field.

### 3.3 NETWORK INFERENCE

Our goal is to use an appropriate geometric representation for the underlying objects in the scene so that we can apply geometric deep learning methods to infer the acoustic scattering field from the object shape. It is also important that our approach should be able to handle highly dynamic scenes with general objects (i.e., no assumption on their convexity or genus). Moreover, our approach can still produce plausible ASFs results for changing typologies. The input to our system will be the point cloud representation ($N \times 3$ matrix, $N$ being the number of 3D points) of an object. We represent each point and its local surface by a higher dimension implicit surface in the latent space formed by an implicit surface encoder. [Thanks to Kolmogorov–Arnold representation theorem, we can represent the pressure field function as a function defined on a set of points ($N \times 3$), which is a multivariate continuous function composed of continuous functions of one variable encoded in a higher dimension. Moreover, borrowing the power of the **De Finetti Theorem**, our pressure field function is further formulated as Eq. 8. Please refer to § 4.4 for more details of our design.]The desired output is the spherical harmonic coefficients vector up to order $L = 3$ described in § 3.2. In practice, acoustic scattering corresponding to different frequencies can exhibit different distributions. Therefore, we train several networks for different frequency bands and in this paper we present the results for $125Hz \sim 1000Hz$. However, this range is not a limit of our approach, as we can handle a wider frequency range, though the complexity of computing the training data can increase as a cubic function of the frequency. We present a novel network architecture to compute these characteristics for arbitrary or deforming objects.

## 4 NETWORK DESIGN AND PROPERTIES

In this section we introduce our shape Laplacian based point representation using implicit surface functions. For predicting ASF, [the detailed geometric characteristics of object representation play an essential role with respect to the simulation frequency.]This motivates our design in terms of differential coordinates (Sorkine, 2006; Do Carmo, 2016), which describe the discretization of the curvilinear integral around a given point. Moreover, we also provide the proof of its permutation invariant property via Newton's identities in the appendix, [as that is important in the context of our sound propagation algorithm]. Finally, following de Finetti theorem (Heath & Sudderth, 1976), we justify our design of implicit surface functions and [show its benefits in terms of accurately computing the acoustic scattering functions for arbitrary objects].

### 4.1 LOCAL SURFACE SHAPE AND IMPLICIT SURFACE ENCODER

Previous works on point cloud learning algorithms mostly focus on designing per-point operations (Charles et al., 2017), encoding per-point features to estimate continuous shape functions (Park et al., 2019; Xu et al., 2019), or minimizing loss between a point normal vector and its connected vertices (Liu et al., 2019). We extend these ideas to compute the acoustic scattering functions.

[R3Q1: For each point in the input cloud and its neighborhood in the Euclidean space, we assume that it can form a piecewise smooth surface around the point.]Each point is encoded by the shared multi-layer perceptron (MLP) (Rumelhart et al., 1985) and can be represented by a vector in the higher dimensional latent space $\mathcal{Z} \in \mathcal{R}^{128}$(see Figure 2) [R3Q1: Thus, a piecewise-linear approximation of the surface around a given point can be used to estimate the local surface shape, where the differential coordinate] (Sorkine, 2006; Do Carmo, 2016) (i.e., $\delta-$ coordinates) of each vertex $v_i$ can be expressed as:

$$\vec{\delta_i} = \frac{1}{d_i}\Sigma_{j\in\mathcal{N}(i)}(\vec{v_i} - \vec{v_j}).$$

(4)

Here $\delta_i$ encapsulates the local surface shape, $\mathcal{N}(i)$ represents the $K$ nearest neighbors of vertex $v_i$ in the Euclidean space, and $d_i = |\mathcal{N}(i)|$ is the number of immediate neighbors of $v_i$. To estimate

the mean curvature of the local surfaces formed by each point and its spatial neighbors, we use the radial basis function (RBF) $\varphi(\cdot) = \exp^{-||\cdot||^2}$ to weigh each vector, rather than using the uniform weight shown in Equation (4). Since there are $N!$ permutations for a point cloud with $N$ vertices, every operation on point clouds should be permutation invariant (i.e., the permutation of input points should not change the output of our network). [Therefore, we design an algorithm based on discrete laplacian and prove its permutation invariance property in § 4.4. ]Our weight function is designed to be positive definite and symmetric for any choice of data coordinates.

## 4.2    [R1Q4,R3Q3-4:IMPLICIT SURFACES AND DISCRETE LAPLACIAN]

To encapsulate the local surface shape, each point $v_i$ is projected onto a higher dimensional space using MLPs, and the implicit surface is defined on the latent space $z$ as shown on the top of Figure 2. Specifically, for one layer MLP, the representation is $z_i = relu(\overrightarrow{w} \cdot \overrightarrow{\delta_i} + b)$, where $\overrightarrow{w}$ and $b$ are learnable parameters in our network. To calculate the $\delta-$ coordinates of a point $v_i$ and the closed simple surface curve around it, its immediate neighbors in the Euclidean space (i.e., $\mathcal{N}(v_i)$) is used to evaluate Equation (5).

$$\overrightarrow{\delta_i^{implicit}}(v_i) = \Sigma_{j\in\mathcal{N}_{(i)}} \frac{\varphi(\overrightarrow{v_i} - \overrightarrow{v_j})(\overrightarrow{v_i} - \overrightarrow{v_j})}{\Sigma_{j\in\mathcal{N}_{(i)}}\varphi(\overrightarrow{v_i} - \overrightarrow{v_j})}. \tag{5}$$

In the latent space, the local surface shape is encoded as an implicit surface where $\mathcal{N}(z_i)$ is evaluated. The direction of the differential coordinate vector, also defined in Equation (5), approximates the local normal direction. Following Taubin (1995), the discrete Laplacian of a point $v_i$ is given by the weighted average over the neighborhood is represented as:

To compare the two $\delta-$ coordinate representations, we highlight the error in decibel (dB) between the pressure fields reconstructed from groundtruth spherical harmonics term and the predicted ones using different neural networks in Table 2. We observe that $\delta^{implicit}-$ coordinates result in smaller errors. This indicates that our formulation provides a better approximation of ASFs.

## 4.3    [R1Q6-7, R3Q3-4: NEURAL NETWORK DESIGN]

Our neural network takes the point cloud as an $N \times 3$ input where $N$ represents the number of points in the point cloud. The output is the spherical harmonic coefficients $c_l^m, -l \leq m \leq l, 0 \leq l \leq 3$, resulting a vector of length 16. The network is illustrated in Figure 2.

[ For each point $v_i$ in the $N \times 3$ point cloud, the discrete laplacian is evaluated on its immediate neighbors $\mathcal{N}(v_i)$ according to Equation (5). Then one layer of CNN and three layers of MLP are used to encode the piecewise-smooth local shape around $v_i$ into a latent space $\mathcal{Z} \subset \mathcal{R}^{128}$. For each $z_i \in \mathcal{Z}$, the discrete laplacian Equation (5) is further evaluated where $\mathcal{N}(v_i)$ (i.e. neighbors of $v_i$ in the Euclidean space $\mathcal{R}^3$) and $\mathcal{N}(z_i)$ (i.e. neighbors of $z_i$ in the latent space $\mathcal{Z}^{128}$) scales the $\delta-$ coordinates $\mathcal{R}^{128}$, respectively, yielding a $2K \times 128$ matrix. The final representation of $v_i$ is shown below: ]

$$\begin{aligned}feature(v_i) = \Bigg( &z_i^T, \frac{\varphi(\overrightarrow{v_i} - \overrightarrow{v_1})(\overrightarrow{z_i} - \overrightarrow{z_1})^T}{\Sigma_{j\in\mathcal{N}(v_i)}\varphi(\overrightarrow{v_i} - \overrightarrow{v_j})}, \dots, \frac{\varphi(\overrightarrow{v_i} - \overrightarrow{v_K})(\overrightarrow{z_i} - \overrightarrow{z_K})^T}{\Sigma_{j\in\mathcal{N}(v_i)}\varphi(\overrightarrow{v_i} - \overrightarrow{v_j})}, \\ &\frac{\varphi(\overrightarrow{z_i} - \overrightarrow{z_1})(\overrightarrow{z_i} - \overrightarrow{z_1})^T}{\Sigma_{j\in\mathcal{N}(z_i)}\varphi(\overrightarrow{z_i} - \overrightarrow{z_j})}, \dots, \frac{\varphi(\overrightarrow{z_i} - \overrightarrow{z_K})(\overrightarrow{z_i} - \overrightarrow{z_K})^T}{\Sigma_{j\in\mathcal{N}(z_i)}\varphi(\overrightarrow{z_i} - \overrightarrow{z_j})} \Bigg),\end{aligned} \tag{6}$$

where $K = 5$ is the number of the nearest neighbors and 128 is the dimension of the latent space. Now we have the piecewise approximation of the local shape around $v_i$ as described in Equation 6, which is composed of the center point $z_i \in \mathcal{R}^{128}$ and the $2K \times 128$ weighted discrete laplacian matrix. The vectors are concatenated and convolved with kernels of size $[1, (1 + 2K)]$, followed by two layers $MLP$ and four fully connected layers with *tanh* activation functions since the value of $c_l^m$ ranges from $[-1, 1]$. The output of the network is the spherical harmonics coefficients $c_l^m$ and the loss function is the $\mathcal{L}2-$ norm of the difference between the predicted $c_l^m$ and the ground truth calculated using *FastBEM Acoustics solver* [1].

---

[1]https://www.fastbem.com/

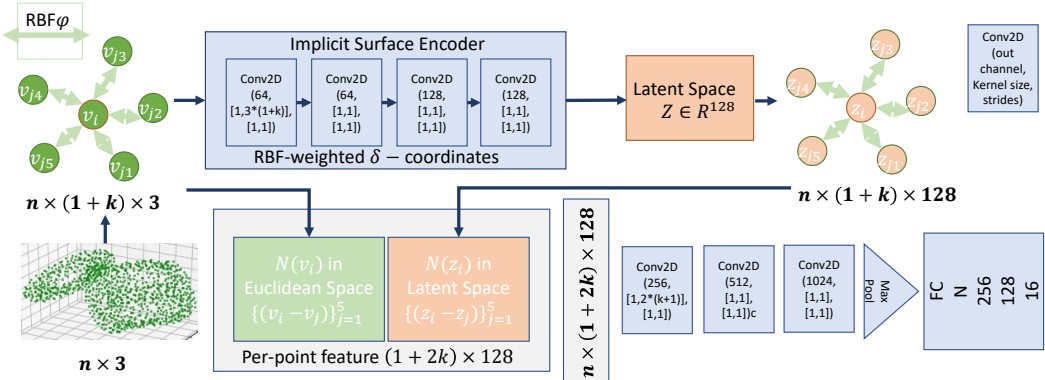

Figure 2: **Network architecture:** The input of our neural network is a $N \times 3$ point cloud ($N = 1024$) and the output of our network is the spherical harmonic coefficients as a vector of length 16 (up to $3^{rd}$ order spherical harmonics). [R1Q6-7, R3Q3-4 : We encode the piecewise smooth local shape around every point in the point cloud, forming a high dimensional latent space in $\mathcal{R}^{128}$. The $\delta-$ coordinates where neighbors are found in the Euclidean space and the latent space are concatenated to represent the per-point feature. We leverage the geometric details by several Conv2D and MLP layers and output 16 spherical harmonics coefficients $c_l^m$.]

## 4.4 [R1Q2-5, R2: DISCRETE LAPLACIAN AND PERMUTATION INVARIANCE]

In this section we provide the proof of the permutation invariant property of the RBF-weighted discrete Laplacian Equation (6). We only discuss the function composed of vectors in the high dimensional latent space projected by the implicit surface encoder shown on the top of Figure 2.

Let $f_i(v_1, \ldots, v_N) = \Sigma_{j \in N(v_i)} \psi(v_i - v_j)$, where $\psi$ is a continuous function composed of $MLP$ layers. According to **Kolmogorov–Arnold representation theorem** (or superposition theorem) and its relationship to multi-layer neural networks (Kůrková, 1992), which states that every multivariate continuous function can be represented as a superposition of continuous functions of one variable, $f_i(v_1, \cdots, v_N)$ can rewritten as: $\Phi\big(\Sigma_{j=1}^N \Psi(v_j)\big)$, where $\Psi : \mathcal{R}^3 \to \mathcal{R}^{2N+1}$ and $\Phi : \mathcal{R}^{2N+1} \to \mathcal{R}^3$. In this paper, we do not project $v$ onto $\mathcal{R}^{2N+1}$; instead, we encode the $v$ as $z \in \mathcal{R}^{128}$. First of all, we show that $f_i$ can be regarded as a sum-of-power mapping defined by the coordinate functions. Second, we show that such mapping is **injective** and has **continuous inverse mapping**. The domain of $f_i$ is a compact set, and thereby the image of $f_i$ is a compact set as well. Thus $f_i$ is homeomorphism between its domain and its image. Further, we demonstrate that $f_i$ is a **permutation invariant continuous** function since it can be considered as compositions of sum-of-power mappings.

Consider $\Psi(v) := [1, v, v^2, \cdots, v^M], \Psi : \mathbb{R}^3 \to \mathbb{R}^{M+1}$, where $M$ is the dimension of our implicit surface encoder space. Let $E(v) = \Sigma_{j=1}^M \Psi(v_j)$

$$
\begin{aligned}
E(v) &= \Big[M, \sum_{m=1}^M v_m, \sum_{m=1}^M v_m^2, \cdots, \sum_{m=1}^M v_m^M\Big] \\
&= [M, p_1(v_1, v_2, \cdots, v_M), p_2(v_1, v_2, \cdots, v_M), \cdots, p_M(v_1, v_2, \cdots, v_M)], \\
&= [E_0(v_1, v_2, \cdots, v_M), E_1(v_1, v_2, \cdots, v_M), E_2(v_1, v_2, \cdots, v_M), \cdots, E_M(v_1, v_2, \cdots, v_M)],
\end{aligned}
\tag{7}
$$

where $p$ is the *power-sums* (defined in Equation (12), Appendix A) and we can observe that $p$ is symmetric (i.e., does not change if we permute $v_1, \cdots, v_M$) and homogeneous of degree $M$. $E_q$ is defined in Lemma 4.1 (Appendix A). According to Lemma 4.1, each element in $E(v)$ is injective. Then by Lemma A.2, $E(v)$ has inverse continuous mapping. Therefore, $E$ is homeomorphism between $\mathcal{R}^3$ and $\mathcal{R}^{M+1}$. Since $\Phi$ is the composition of several continuous functions $\psi$, the continuity of $\Phi$ holds as well.

**Lemma 4.1.** *Let* $\mathbb{X} = \{(x_1, \cdots, x_M) \in [0,1]^M : x_1 \leq x_2 \leq \cdots \leq x_M\}$. *The sum-of-power mapping* $\mathbb{E} : \mathbb{X} \to \mathbb{R}^{M+1}$ *defined by the coordinate functions* $:E_q(X) := \sum_{m=1}^M (x_m)^q, q = 0, \cdots, M$ *is injective.*

The proof of the lemma is in Appendix $A$. We can conclude that there are $\Phi$ and $\Psi$ such that $f_i$ is permutation invariant as long as $\Psi$ and $\Phi$ are continuous functions. To compute the acoustic scattering functions, we further introduce the De Finetti Theorem and exhibit the connection between our design of Discrete Laplacian based implicit surface functions and pressure field approximation.

### 4.5 DE FINETTI THEOREM AND IMPLICIT SURFACE FUNCTIONS

Our idea is to approximate the *pressure field* defined by a set of points. And we borrow power from the following theorem (Heath & Sudderth, 1976):

**Theorem 4.2** (De Finetti Theorem). *A sequence* $(x_1, x_2, \ldots, x_n)$ *is infinitely exchangeable iff*

$$p(x_1, \ldots, x_n) = \int \prod_{i=1}^{n} p(x_i|\theta)P(d\theta),\tag{8}$$

*for some measure $P$ on $\theta$.*

Thanks to the continuity nature of neural networks, we replace $P(d\theta)$ in Theorem (8) with a more general form $P(\theta)d\theta$ since we assume that the underlying latent layer is *absolute continuous* according to Radon–Nikodym theorem (i.e., the distribution on $\theta$ has density. [R1Q5: Note that some singular measures such as Dirac delta *does not have density (Radon–Nikodym derivative)*. For example, each Radon measure (including Lebesgue measure , Haar measure, Dirac measure) can be decomposed into one absolutely continuous measure and one mutually singular measure.]). Then we derive that the acoustic scattering function of a given set of points $\mathcal{X} = \{x_1, x_2, ..., x_n\}$ and the underlying latent space $\mathcal{Z}$ can be formulated in the form:

$$f_{scatter}(\mathcal{X}) = \int dz \int \prod_{i=1}^{n} p(x_i|\theta)P(\theta|z)d\theta.\tag{9}$$

[Since our latent space vector $z$ can be regarded as compositions of sum-of-power mappings (Equation (12), Appendix A), the output pressure field of our network is permutation invariant.]

## 5 EVALUATION

### 5.1 DATA GENERATION AND TRAINING

We need a sufficiently large dataset for training and evaluating our method. We sample 100,000 3D objects from the *ABC Dataset* (Koch et al., 2019). All mesh files are pre-processed to be randomly rotated, scaled between $1 \sim 2m$ and centered. The *FastBEM Acoustics* [2] is used to generate accurate acoustic scattering fields. Specifically, we simulate a plane wave travelling to the $-x$ axis direction, and all objects are assumed to have a zero Neumann boundary condition to solve Equation (2). While other boundary conditions are possible, we do not extensively study this variable in this work. After the acoustic scattering field has been solved at frequencies bands of $\{125, 250, 500, 1000\}Hz$, we use the *pyshtools* [3] package (Wieczorek & Meschede, 2018) to project the directional field to spherical harmonic coefficients $c_l^m$. By using a maximum order of $L = 3$, we are able to maintain a relative projection error below $2\%$. The original object meshes are also converted to point clouds using furthest sampling for 1024 points. The data generation pipeline takes about 12 days using a desktop with 32 Intel(R) Xeon(R) Gold 5218 CPUs.

We split our dataset into training, validation and test set following a $8:1:1$ ratio. Our neural network is trained on a NVIDIA GeForce RTX 2080 Ti GPU and takes less than 1ms per object for inference. The Adam algorithm is used in optimizer and it decays exponentially with decay rate and decay step equal to $0.9$ and $10\times$ #training examples, respectively. Our code and data will be released upon publication.

### 5.2 RESULTS AND COMPARISONS

We evaluate our network performance on unseen objects from the *ABC dataset*, perform ablation study on RBF-weighted $\delta-$ coordinates and implicit surface encoders, and show that our network

---

[2] https://www.fastbem.com/
[3] https://shtools.oca.eu/shtools/public/index.html

is robust to Gaussian noise ($\sim N(0, 0.05)$). We compare our results with PointNet (Charles et al., 2017) and DGCNN (Wang et al., 2019) in Table 2 and Table 3. We justify the design of our network with this ablation study, including the use of $\delta-$ coordinates, RBF-weighted function as well as the implicit surface encoder, as highlighted in Figure 2. We use PointNet, where only per-point MLP layers are applied on each point in the point cloud, and DGCNN, which dynamically build graphs according to $K$ nearest neighbors, as the baseline. RBF-weighted $\delta-$ coordinates (as described in Equation (5)), uniformly weighted $\delta-$ coordinates (described in Equation (4)), and implicit surface encoder shown at the top of Figure 2 are considered as subjects of ablation studies. We observe that our fine-grained geometric feature representation in Equation (6) results in larger reduction in dB error. Moreover, our discrete Laplacian based representation is more robust to the Gaussian noise in the test set compared to DGCNN.

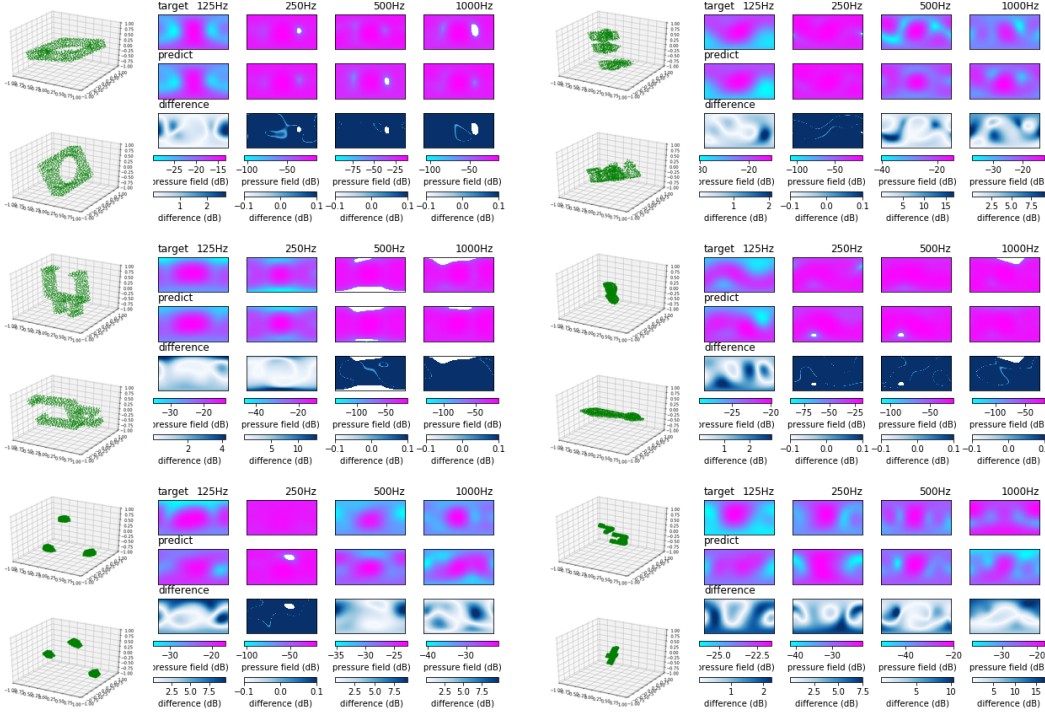

Figure 3: **ASF prediction and groundtruth comparison:** We expand the predicted spherical harmonics coefficients onto a latitude-longitude (lat-long) map, representing the directional distribution of the scattering field. We randomly choose 6 point clouds from the test dataset, which have not been seen during training. Each point cloud is visualized from two viewing angles, followed by their groundtruth/target, predicted, and error/difference lat-long maps in four frequency bands. We see a good match between our predicted fields and the groundtruth fields.

## 5.3 APPLICATION IN INTERACTIVE SOUND RENDERING

[R1Q1: Our geometric deep learning scheme facilitates a wide range of applications related to wave acoustics simulation. Pre-computed acoustic transfer (James et al., 2006; Wang & James, 2019), wave acoustics perceptual encoding (Raghuvanshi et al., 2016) and wave-ray sound propagation (Yeh et al., 2013), which heavily rely on extensive simulation using accurate wave solvers, can all benefit from our efficient learning-based approach. Our method utilizes the connection between an object's shape and its scattering property to remove the pre-computation overhead in existing wave acoustics rendering methods. Our additional spherical harmonics encoding makes it suitable to be integrated into most wave-ray hybrid sound rendering engines.

To illustrate one such application, we integrate our predicted ASFs into a state-of-the-art ray-tracing based acoustic simulator (Schissler & Manocha, 2017). The ray-tracing based simulator assumes all sound waves travel as rays. This works well for high-frequency sounds but do not hold for low-

Table 2: **Ablation study:** In this evaluation, we compare the performance on uniform $\delta-$ coordinate in Eq. (4), weighted $\delta-$ coordinates in Eq. (5) and implicit surface estimation in Eq. (6) on our test dataset (including $10k$ objects) at frequency bands {125Hz, 250Hz, 500Hz, 1000Hz}. The best result for each frequency is highlighted in **bold** (lower error is better). We alter between choosing Eq. (4) and Eq. (5) that results in four different combinations: Row 1 and 2, 3 and 4. Next, we experiment the use of implicit surface encoder (Row 3 and 4). Our proposed network design (Row 4) shows superior performance in terms of ASF approximation for all four frequencies. A lower value indicates a better result.

| Methods {RBF-weighted $\delta$-coord — implicit surface} | Error in dB | | | | #Params | Inference Time sec. per 10000frames |
|---|---|---|---|---|---|---|
| | 125Hz | 250Hz | 500Hz | 1000Hz | | |
| {✗— ✗} | 3.49 | 3.56 | 3.71 | 4.23 | 30k | x |
| {✓— ✗} | 3.38 | 3.41 | 3.57 | 4.47 | 30k | x |
| {✗— ✓} | 3.28 | 3.38 | 3.52 | 3.85 | 30k | x |
| {✓— ✓} (ours) | **2.98** | **3.06** | **3.22** | **3.76** | 34k | 47.10 |
| PointNet (Charles et al., 2017) | 3.66 | 3.44 | 3.43 | 4.16 | 80k | 8.95 |
| DGCNN (Wang et al., 2019) | 3.71 | 3.56 | 3.59 | 4.21 | 103k | 26.80 |

Table 3: **Robustness test**: In this robustness test, we add i.i.d. noise to every point in the point cloud, which follows $N(0, 0.05)$, to the test set. Note that DGCNN finds $K-$ neighbors at each iteration as we did. However, our algorithm is more robust to the noisy data input thanks to the design of the shape Laplacian based point representation.

| Methods | Error in dB | | | |
|---|---|---|---|---|
| | 125Hz | 250Hz | 500Hz | 1000Hz |
| PointNet | 5.88 | 5.50 | 6.02 | 8.37 |
| DGCNN | 7.91 | 6.65 | 7.31 | 11.32 |
| Ours | **5.20** | **5.09** | **4.70** | **5.51** |

frequency sounds. Instead, our goal is to use our wave-based simulator to better capture the sound propagation effects corresponding to low frequency sounds in dynamic scenes. Therefore, we have simulated wave acoustics up to $1000Hz$ to compensate the low-frequency effects like sound diffraction using our learning method and compute the acoustic scattering functions of each object in the scene. This choice of the simulation frequency of $1000Hz$ is similar to Mehra et al. (2013), Raghuvanshi & Snyder (2018), Rungta et al. (2018) and Morales et al. (2019), as they can provide plausible effects for games, virtual reality, and computer-aided design applications.

The specific wave-ray coupling scheme we use resembles Rungta et al. (2018) but can be switched to other compatible rendering engines. We use our trained network at runtime to predict ASFs of static and dynamic objects, and evaluate the spherical harmonic expansion in Equation (3). Then the sound is rendered at given listener locations. A summary of details is also provided in Appendix B. We demonstrate the final audio-visual rendering in our supplemental video. ]

## 6 Conclusion and Limitations

We present a novel geometric deep learning approach that infers the 3D acoustic scattering field induced by an object from its shape. Our algorithm uses discrete-laplacian based implicit function encoders to better capture the geometric properties of an object and achieves lower error compared with existing general point-based frameworks. Our shape and sound field representation are well-suited for applications that require real-time wave acoustics characteristics.

Currently we study the case of sound-hard surfaces and a limited range of simulation frequencies, but these are not an inherent limitation of our method. The simulation time needed to create a large training set can grow as a cubic function of the sound frequency. Incorporating more types of sound surfaces can further increase the training overhead of all geometric deep learning methods, including ours. In the future, we propose to extend these methods to real-world objects and use them for localization and non-line-of-sight computations.

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

# Appendices

## A  PERMUTATION INVARIANCE

In this section we provide the proof of permutation invariant property of the RBF-weighted discrete Laplacian Equation (6) thanks to Newton's identity. First of all, Sec. A.1 serves as a recap of Newton's identity, which is applied in the proof of Lemma 4.1. Lemma A.2 shows that the *sum-of-power mapping* defined in Lemma 4.1 has continuous inverse mapping. Both of the lemmas provide the background for the permutation invariance of our Discrete Laplacian in Equation (6). Furthermore, we demonstrate with the De Finetti Theorem that the acoustic scattering functions can be approximated using the underlying latent space.

### A.1  NEWTON'S IDENTITIES

Newton's identities formulas, also known as **theory of equations**, build bridge between elementary symmetric polynomials and power sums. Consider a polynomial f of degree n with roots $x_1, \cdots, x_n$. Assume that $s_0 = 1$, $f$ is monic.

$$
\begin{aligned}
f(x) &= s_0 x^n + \cdots + s_{n-1} x + s_n \\
&= \prod_{i=1}^{n} (x - x_i) \\
s_r &= (-1)^r \sum_{j_1 < \cdots < j_r} x_{j_1} \cdots x_{j_r} \\
s_r' &= s_r \cdot (-1)^r
\end{aligned}
\tag{10}
$$

Note that $s_r$ and $s_r'$ are **symmetric** (i.e. does not change if we permute $x_1, \cdots, x_n$) and **homogeneous** of degree $r$.

These $s_r'$ polynomials, often called **elementary symmetric polynomials**, form a bases for all symmetric polynomials.

For example,

$$
\begin{aligned}
s_1 &= (-1) * (x_1 + \cdots + x_n) \\
s_2 &= x_1 x_2 + x_1 x_3 + x_1 x_4 + \cdots \\
s_3 &= (-1) * (x_1 x_2 x_3 + x_1 x_2 x_4 + x_1 x_2 x_5 + \cdots) \\
&\vdots \\
s_n &= (-1)^n * (x_1 x_2 x_3 \cdots x_n)
\end{aligned}
\tag{11}
$$

Another family of symmetric polynomials are **power sums**, which form a basis for the space of all symmetric polynomials:

$$
p_r(x_1, \cdots, x_n) = x_1^r + \cdots + x_n^r
\tag{12}
$$

The transition formulas between power sums and elementary symmetric polynomials are called **Newton's identities**.

**Lemma 4.1** Let $\mathbb{X} = \{(x_1, \cdots, x_M) \in [0,1]^M : x_1 \leq x_2 \leq \cdots \leq x_M\}$. The sum-of-power mapping $\mathbb{E} : \mathbb{X} \to \mathbb{R}^{M+1}$ defined by the coordinate functions :

$$
E_q(X) := \sum_{m=1}^{M} (x_m)^q, q = 0, \cdots, M
\tag{13}
$$

is injective.

**Proof:** For some $u, v \in \mathbb{X}$, $E(u) = E(v)$, show $u = v$.

Let $u, v$ be the roots of polynomial of degree $M$, we have:

$$
\begin{aligned}
f_u(x) &= \prod_{m=1}^{M} (x - u_m) \\
&= x^M + a_1 x^{M-1} + \cdots + a_{M-1} x + a_M \\
f_v(x) &= \prod_{m=1}^{M} (x - v_m) \\
&= x^M + b_1 x^{M-1} + \cdots + b_{M-1} x + b_M
\end{aligned}
\tag{14}
$$

$a, b$ are elementary symmetric polynomials as described in Equation (10). Elementary polynomials(i.e. $a, b$) can be expressed by power sums.

$$
\begin{aligned}
a_M &= \frac{1}{M!} \begin{vmatrix} E_1(u) & 1 & 0 & 0 & \cdots & 0 \\ E_2(u) & E_1(u) & 2 & 0 & \cdots & 0 \\ E_{M-1}(u) & E_{M-2}(u) & E_{M-3}(u) & E_{M-4}(u) & \cdots & M-1 \\ E_M(u) & E_{M-1}(u) & E_{M-2}(u) & E_{M-3}(u) & \cdots & E_1(u) \end{vmatrix} \\
b_M &= \frac{1}{M!} \begin{vmatrix} E_1(v) & 1 & 0 & 0 & \cdots & 0 \\ E_2(v) & E_1(v) & 2 & 0 & \cdots & 0 \\ E_{M-1}(v) & E_{M-2}(v) & E_{M-3}(v) & E_{M-4}(v) & \cdots & M-1 \\ E_M(v) & E_{M-1}(v) & E_{M-2}(v) & E_{M-3}(v) & \cdots & E_1(v) \end{vmatrix}
\end{aligned}
\tag{15}
$$

$E(u) = E(v)$ implies that $a = b$, and therefore $u = v$.

**Theorem A.1.** *The function $f : \mathbb{C}^M \to \mathbb{C}^M$, $f(c) = x^M + c_1 x^{M-1} + \cdots + (-1)^{M-1} c_{M-1} x + (-1)^M c_M, c \in \mathbb{C}^M$, is homeomorphism (Ćurgus & Mascioni, 2006).*

**Lemma A.2.** *The sum-of-power mapping defined in Lemma 4.1 has continuous inverse mapping.*

The domain of $E$ is a compact set, and $E$ is a continuous function. Therefore, the image of $E$ is a compact set. From Lemma 4.1, $a$ is a continuous function of the power sums $E$.

**Goal:** Show continuity of inverse mapping of $E$.

**Proof:** From Theorem A.1, the continuity of roots $u$ depends on the coefficients $a$. W.L.G, show continuity from $a$ to $u$. Due to the nature of homeomorphism, the mapping from $u$ to $a$ as well as its inverse mapping from $a$ to $u$ are both continuous.

## B HYBRID WAVE-RAY SOUND RENDERING

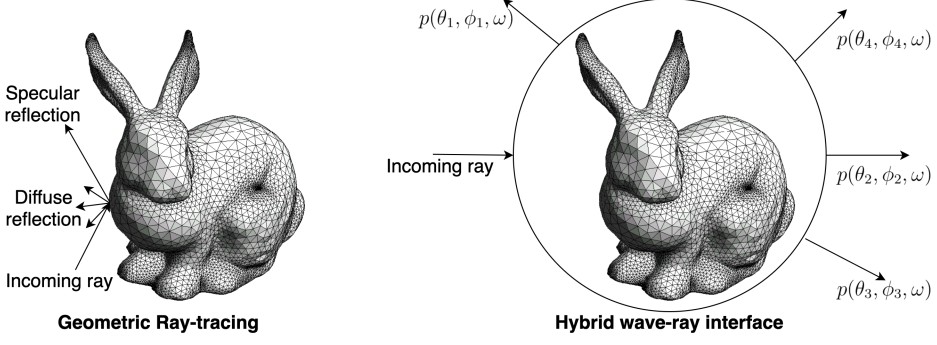

Figure 4: Pure geometric ray-tracing (left) vs hybrid wave-ray coupling (right).

Conventional ray-tracing algorithm finds many ray paths from a sound source to a receiver/listener, and a time delay and energy damping is calculated according to the path length and the order of

reflections of each path. These information can be used to compose an acoustic impulse response for sound rendering in real-time (Kuttruff, 1993).

The core of the ray-tracing process is to sample a large number of reflection directions and their corresponding energy when a ray hits a surface. A pure geometric sound propagation system assumes specular and diffuse reflections (Figure 4 left). In this setup, the reflected rays can only be in directions which have a positive dot product with the normal direction of the hitting point. Therefore, it is impossible for rays to "bypass" an obstacle, while for wave acoustics, such a phenomenon (diffraction) always happens. As an alternative, the acoustic scattering field (ASF) we computed can be used to model diffraction effects. Specifically, when an incoming ray hits a sound scatterer, we sample outgoing directions among all directions, and calculate the energy decay by evaluating the ASF at each direction $(\theta_i, \phi_i)$ for frequency $\omega$ (Figure 4 right). This hybrid wave-ray propagation formulation is general enough to be used with most sound propagation engines.

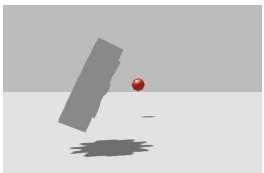 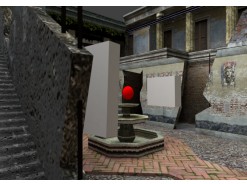 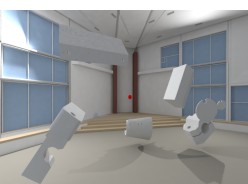 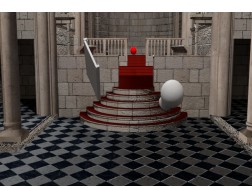

(a) Floor: One static sound scatterer in open space. 10.65ms/frame.
(b) Havana: Two moving walls in half-open space. 6.78ms/frame.
(c) Trinity: Six flying objects in a large indoor room. 12.95ms/frame.
(d) Sibenik: Two disjoint revolving objects in a church. 6.87ms/frame.

Figure 5: Benchmark scenes used for audio-visual rendering in our supplemental video. These are dynamic scences where objects come in close proximity and change topologies. Our learning methods can compute accurate scattering fields for such real-time applications.

We also test sound rendering on four benchmark scenes of different complexity in Figure 5 to perceptually show the smooth and realistic sound rendering results of our method.

