# OpenReview forum: "Fast 3D Acoustic Scattering via Discrete Laplacian Based Implicit Function Encoders"
_ICLR.cc/2021/Conference — Reject_

### Official Review · AnonReviewer2 · 2020-10-28
**Applied geometric deep learning for sound modelling**

**Rating:** 6
**Confidence:** 3

**Review:**

Summary: This paper outlines a geometric deep learning approach for predicting acoustic scattering fields at different frequencies. The fields are represented with spherical harmonic coefficients (only the directional parts), and they use a nearest neighbor approach to construct an implicit surface in a latent space, and leverage a Laplacian in this domain. In their experimental section, they show superior results to other geometric deep-learning architectures that use point clouds, and also perform an ablation study.

Strengths: The results for the method seem to beat out other similar point-cloud-based methods and I appreciate the application to a specific sound-modelling problem.

Weaknesses: I found their description of the geometric encoding part to be sparse on the expository front. Their use of differential coordinates, use of a surface in latent space, and the construction of their feature vector are presented without much discussion of motivation. It would help to compare and contrast with existing point-cloud based methods.

Recommendation: I gave a rating of 6, as I found it difficult to see what insights I should glean from their method. On the other hand, their method seems to work well from a practical standpoint, and has a concrete application that seems useful. For a higher score, I think the paper could benefit from a clearer exposition and a more extensive appendix describing the details of their network architecture. This could include a clearer delineation of the differences between their method and similar geometric deep-learning models.

Typos:
* typologies -> topologies in 3.3
* exponential powers squared in (5)?

---

> ### Author Response · Authors · 2020-11-25
> **Section 4 has been revised to address some potential weakness**
>
> ### R2Q1,  I found their description of the geometric encoding part to be sparse on the expository front. Their use of differential coordinates, use of a surface in latent space, and the construction of their feature vector are presented without much discussion of motivation.
>
> Our approach is inspired by the Discrete Laplacian of graphs, which can provide more detailed geometric characteristics. Discrete Laplcian is inevitable for geometric mesh processing tasks such as shape approximation and compact representation, mesh editing, watermarking, and morphing. We borrow the power from the traditional geometric processing and design our permutation invariant algorithm. Please also refer to the ``Purrpose of EQ(4-6)`  on the top of this page.
>
>
> ### R2Q2, It would help to compare and contrast with existing point-cloud based methods.
>
> -The comparison is shown in Table. 2. We compare with PointNet and DGCNN and both of them are point-cloud based neural networks. PointNet feed MLP into several MLP layers, which only encapsulates the point itself instead of the neighbours around the point; while our method approximates the local surface shape using RBF-weighted discrete laplacians. DGCNN presents EdgeConv, which convolves the $K$ nearest neighbor; while our method takes advantage of discrete laplacian and implicit surface encoder to aggregate more detailed geometric characteristics. For acoustic scattering functions, the detailed geometric characteristics of object representation play an essential role with respect to the simulation frequency. Compared to DGCNN, we have much lower dB error for 1000Hz.

---

### Official Review · AnonReviewer1 · 2020-10-28
**Interesting but narrow scope, technical inconsistencies and lack of details**

**Rating:** 4
**Confidence:** 4

**Review:**

This paper proposes to train a neural network to predict the acoustic scattering effects of a 3D object. The input of the system is a 3D point cloud representing the object, and the outputs are 16 coefficients from the spherical harmonic decomposition of the sound field resulting from an incoming planar wave to the object at a given frequency (4 frequencies are considered, with one specialized network trained for each). The main methodological contribution is a carefully crafted learned representation of the 3D point cloud. Reported results show that the proposed architecture outperform two other architectures while using less parameters and is able to predict sound fields with relatively small errors, using orders of magnitude less computational time than the acoustic simulators used to train the network.

Pros:
- The proposed 3D point cloud representaiton is interesting and potentially applicable to other tasks
- The general idea of using deep learning to accelerate acoustic simulation is interesting
- Reported results and comparisons look reasonnable

Cons:
- The scope of the paper is very narrow, making it of limited interest. The approach is only tested on simple simulations (no real data involved) with rather unrealistic assumptions (zero Neumann boundary, perfect plane wave, SH decomposition to order 3 only) and few frequency bands (125,250,500,1000 Hz), but several days of computations to generate the training data are still reported by the authors. It is not clear if and how the proposed approach would scale to more realistic/useful settings.
- I found sections 4.4, 4.5 and the Appendix cryptic. A context of what exactly is discussed here and how it relates to the proposed method is seriously missing.  Two obscure lemmas and a thorem are packed in a dozen lines of text without any references. I was not able to follow the mathematical soundness of this, or indeed even to understand how it precisely relates to the rest of the paper.
- In general the paper looks like it has been typed in a rush, with some typos but more importantly several technical inconsistencies, making ts soundness hard to verify. One important source of confusion is the radial basis function \phi. Below equation (4) it is defined as exp(-||.||^2). In equation (5) an exponential of norm is used but this time with no square. This is again used in the denominator of the first line of eq. (6), although this is presumably a typo? Then in section 4.4, \phi is defined as a function of completely different nature, from a scalar to a vector of powers, alllegedly "without loss of generality". I do not see how to make sense of all these different definitions of phi. Another issue are the cryptic 4 lines preceding Lemma 4.2, ending with "the distribution on \theta has density" : has density what? Also, below equation (6), the authors write "Equation (6), is further fed into the MLP, forming the differential coordinates in Equation (5)" it is not clear in what sense eq. (5) relates to the MLP or equation (6). Last, details are missing in the description of the neural network architecture: in what precise sense are the MLP weights shared? Over which dimension is the Max pool applied? What non-linearities are used?

Overall, I feel like the flaws of the paper outweights its strength. It would benefit from an expansion to a longer format in order to better explain the theoretical and methodological aspects and extend experiments to more diverse and/or realistic settings.

===== Edit after authors' revisions ======
The authors made some efforts to improve the exposition of the most obscure parts of the paper, but in my opinion this is not sufficient and I am still not able to fully grasp the connection between section 4.4 and 4.5 and the rest of the paper. The added parts (in blue) contain additional typos and, like the rest of the paper, look like they have been type in a rush. For instance:
- "First of all, we show that f_i can be regarded as a sum-of-power mapping defined by the coordinate function" -> I don't see where this is showed in the paper
- "f_i(v_1, · · · , v_N ) can rewritten as" -> can be rewritten as

I'd encourage the authors to resubmit their work using a longer paper format that would allow them to better expose the details of their investigation.

---

> ### Author Response · Authors · 2020-11-25
> **Response for Q1-3, Scope issue, better consistencies and more details reflected in writing**
>
> ### R1Q1, The scope being “narrow”:
> We were not able to provide many applications and benefits of our work due to space limitations. We have added more details to the revised paper and appendix.
> 1) “Simple” simulations: the acoustic scattering field is an intrinsic property of an object and is very hard to measure for real-world objects, as one needs a special and expensive setup, which is not standardized. However, they can be computed for geometric shapes by solving the wave equation using numerical solvers (Section 5.3).  State-of-the-art wave solvers are able to show a good agreement with analytical solutions for a special category of objects (e.g. highly regular & symmetrical shapes).  Moreover, in large scenes with multiple objects, it is not easy to measure or accurately evaluate the global sound pressure field, but the results computed using numeric acoustic solvers can closely match some measurable acoustic metrics like the RT_60 (reverberation time) or DRR (direct-to-reverberation ratio). Therefore, many state-of-the-art audio-visual rendering systems [Wang et al. 2019; Raghuvanshi et al. 2016; Rungta et al. 2018] heavily rely on these accurate numerical solvers. Our approach is motivated by these needs. Our learning methods can compute the ASF for dynamic scenes or deformable objects (see Figure 3). As a result, ours results in an accurate wave-based sound propagation with dynamic or deformable objects (Section 5.3).
>
> 2) Parameter assumptions (perfect plane wave, SH decomposition to order 3 only): In practice, constant sound speed and plane wave assumptions are commonly used in sound propagation methods used for indoor scenes, as mentioned in [Mehra et al. 2013; Mehra et al. 2015; Morales et al. 2015; Raghuvanshi and Snyder 2014; Rungta et al. 2018]. These methods have been validated by comparing their performance with real-world measurements, which justify the use of such assumptions. The order 3 decomposition has also been tested to yield reproduction errors below 2% (see Section 5.1).  Our approach can be easily extended to use higher order SH decompositions. In this work we offer a novel learning algorithm to compute the scattering fields using geometric learning, which highlights the potential of accurate sound propagation in dynamic or deformable models.
> 3) Frequency bands:  We had mentioned (Section 3.3) that “this range is not a limit of our approach, as we can handle a wider frequency range, though the complexity of computing the training data can increase as a cubic function of the frequency”. We refer to [Rungta et al. 2018] for the practical way of incorporating wave acoustics into a real-time sound rendering pipeline by using ray-tracing method to simulate the effects at higher frequencies. Most prior sound propagation algorithms limit the wave-computations around 1000Hz (Section 5.3), as that results in plausible sound effects for interactive applications such as games, virtual reality, computer-aided design.
>
> ### R1Q2,  A context of what exactly is discussed here and how it relates to the proposed method is seriously missing
>
> -We have explained EQ(4-6) above. We prove that our differential coordinate-based point cloud representation is permutation invariant in Lemma 4.1 and Theorem 4.2.
>
> ### R1Q3, Two obscure lemmas and a theorem are packed in a dozen lines of text without any references. I was not able to follow the mathematical soundness of this, or indeed even to understand how it precisely relates to the rest of the paper.
>
> -We have revised Section 4.4 accordingly, hoping to have presented these derivations in a more logical order and connected them better to the paper.

---

> ### Author Response · Authors · 2020-11-25
> **Response for Q4-7**
>
> ### R1Q4,  One important source of confusion is the radial basis function $\phi$. Below equation (4) it is defined as exp(-||.||^2). In equation (5) an exponential of norm is used but this time with no square. This is again used in the denominator of the first line of eq. (6), although this is presumably a typo?
>
> -We are sorry that EQ(5-6) should be revised to be squared. We have changed the notation $\phi$ in Section 4.4 to avoid confusion.
>
> ### R1Q5,Another issue are the cryptic 4 lines preceding Lemma 4.2, ending with "the distribution on $\theta$ has density" : has density what?
>
> -Some singular measures such as Dirac delta do not have density. Each Radon measure can be decomposed into one absolutely continuous measure and one mutually singular measure.
>
> ### R1Q6, Also, below equation (6), the authors write "Equation (6), is further fed into the MLP, forming the differential coordinates in Equation (5)" it is not clear in what sense eq. (5) relates to the MLP or equation (6).
>
> -For each point in the $(N \times 3)$ point cloud, the discrete laplacian is evaluated on its immediate neighbors (i.e. $N(v)$)  according to Eq. 4. Moreover, we apply the radial basis function to scale the direction of the normal vector, as shown in Eq. 5.
> Then one layer of CNN and three layers of MLP are used to encode the piecewise-smooth local shape around the point into a latent space ($Z \in R^{128}$). For each z, the discrete laplacian Eq. 5  is further evaluated where $N(v)$  (i.e. neighbors of v in the Euclidean space $R^3$ and $N(z)$  (i.e. neighbors of $z$ in the latent space $Z$) scales the delta-coordinates, respectively, yielding a $[2K \times 128]$ matrix. The final representation of $v$ is shown as Eq. 6.
>
>
> ### R1Q7, Over which dimension is the Max pool applied? What non-linearities are used?
>
> -The dimension of the per-point feature is $(1 + 2 \times k) \times 128$, and the dimension of the last MLP is 1024, therefore the max pool is applied on a $1024 \times 1024$ matrix. The non-linearities are introduced by tanh since SH coefficients range from -1 to 1.

---

### Official Review · AnonReviewer3 · 2020-10-30
**Impressive results but method is not properly explained**

**Rating:** 3
**Confidence:** 3

**Review:**

This work introduces a new architecture for predicting the acoustic scattering fields from an object and an incident plane wave. This is done by describing the object using a point cloud mesh whose local geometric properties is encoded in a set of latent space vectors and processed by a deep neural network. The resulting architecture can predict the acoustic scattering fields quickly with high accuracy for a given test set.

The problem formulated is an interesting one and the numerical results are very impressive. The description of the method, leaves quite a bit to be desired. The details of the network are described in vague terms and when formulas are introduced, much of the notation is not defined or not well motivated. As a result, it becomes very hard to understand exactly how and why the proposed method works. For this reason, I do not recommend that this work be published as part of the proceedings.

As stated above, the numerical results are impressive. The authors state that their network is able to calculate the acoustic scattering field in less than one millisecond for a given object. It is not clear how this compares to the baseline method or the state of the art (PointNet and DGCNN). Section 5.1 mentions it taking twelve days to generate the ground truth data set of 100000 objects, which comes out to about 10 seconds, so this would be a considerable speedup. Still, it is not clear how much faster the proposed method is compared to the state of the art.

The main problem is the description of the method, starting in eq. (4), where “differential coordinates” are defined. These seem to be vectors in R³, but what is their purpose? What do they encode? Then a variant of this is introduced in (5), where a learned latent space is used in the form of the z coordinates. How exactly are these latent space coordinates learned? What is the objective? Then the vector in eq. (6) is introduced. It is sometimes referred to as a discrete Laplacian, but it is not clear how (in the equation, it is simply labeled “feature”). Sections 4.4 and 4.5 are similarly impenetrable. Furthermore Section 4.5 refers to Appendix B, which defines a theorem, but which has no proof.

---

> ### Author Response · Authors · 2020-11-25
> **We have re-organized Section 4 to clarify our results**
>
> ### R3Q1:The main problem is the description of the method, starting in eq. (4), where “differential coordinates” are defined. These seem to be vectors in R³, but what is their purpose?
>
> -For each point in the input point cloud and its neighborhood in the Euclidean space, we assume that its local neighborhood is a piecewise smooth surface around the point. For example, the piecewise-linear approximation of the surface around a given point(i.e. EQ(4)) can be used to estimate the local surface shape [Sorkine, 2006].
>
> ### R3Q2: It is not clear how this compares to the baseline method or the state of the art (PointNet and DGCNN)
>
> -We use public implementations of  PointNet (https://github.com/charlesq34/pointnet) and DGCNN(https://github.com/WangYueFt/dgcnn) to compare the performance of our novel network. We did not change the underlying network architectures of PointNet and DGCNN, instead only changed the last layer to be the fully connected layer that outputs 16 SH coefficients (see Section 3.2).  Our method exhibits higher accuracy in terms of dB error (see Table 2),  while the number of training parameters used by our approach is less than PointNet and DGCNN (also in table 2). The inference time (second per 10624 frames) for PointNet, DGCNN, and our method are (8.95, 26.80, 47.10)s, respectively. The conclusion is that all these networks are able to achieve per object inference rates that will work for interactive applications thanks to the efficiency of GPU operations, while traditional wave solvers will take at least thousands of times longer which can only be run off-line.
>
> ### R3Q3:Then a variant of this is introduced in (5), where a learned latent space is used in the form of the z coordinates. How exactly are these latent space coordinates learned? What is the objective?
>
> -The network of the implicit surface encoder, displayed on the top of  Fig. 2, is composed of one Conv2D and three MLP layers.  We have revised Figure 2 in the paper. We also changed the description and the caption to explain these issues in Sections 4.2 and 4.3.
>
> ### R3Q4:Then the vector in eq. (6) is introduced. It is sometimes referred to as a discrete Laplacian, but it is not clear how (in the equation, it is simply labeled “feature”).
>
> -Eq. 4 is the most basic variant of discrete Laplacian of graphs. We can consider each point in the point cloud and its immediate neighbours compose a local shape. Eq. 6 is composed of the vectors in Eq. 5 evaluated using both $N(v)$ and $N(z)$. We have Figure 2 in the paper and revised the description.

---

### Author Response · Authors · 2020-11-25
**Authors' Response : Sec 4 & 5 & Appendix have been revised**

We thank all of the reviewers for your detailed feedback and constructive suggestions. Due to space limitations, we were unable to present many details about our geometric learning methods and applications to sound propagation. We have revised the paper (mostly Section 4) and the rebuttal to address these comments. The changed texts are highlighted in cyan in the revised submission in response to the summarized questions from all reviewers below.

### Purpose of Eq (4-6) in our paper
The complexity of sound propagation algorithms increases as a cubic function of the maximum simulation frequency (as shown in Eq. 1). In order to simulate higher frequency (i.e., lower wavelength), we need a more detailed geometric representation of the objects and the size of triangles and edge is a function of the wavelength.
Therefore,  for each point in the ($N \times 3$) point cloud, the discrete laplacian is evaluated on its immediate neighbors (i.e. $N(v)$)  according to Eq. 4. Moreover, we apply the radial basis function to scale the direction of the normal vector, as shown in Eq. 5.
Then one layer of CNN and three layers of MLP are used to encode the piecewise-smooth local shape around the point into a latent space ($Z \in R^{128}$) . For each z, the discrete laplacian Eq. 5  is further evaluated where $N(v)$  (i.e. neighbors of v in the Euclidean space $R^3$ and $N(z)$  (i.e. neighbors of $z$ in the latent space $Z$) scales the delta-coordinates, respectively, yielding a $[2K \times 128]$ matrix. The final representation of $v$ is shown as Eq. 6.

---

### Decision · Program_Chairs · 2021-01-07
**Final Decision**

**Decision:**

Reject

**Comment:**

The authors were responsive to the comments of the reviewers, both in
the rebuttal and in the revision to the manuscript.  However, the
reviewers were still concerned about the lack of clarity of the
manuscript, the motivation for the design decisions, and errors,
present also in the rebuttal revisions.